# Enhanced and Tunable Electrorheological Capability using Surface Initiated Atom Transfer Radical Polymerization Modification with Simultaneous Reduction of the Graphene Oxide by Silyl-Based Polymer Grafting

**DOI:** 10.3390/nano9020308

**Published:** 2019-02-24

**Authors:** Erika Kutalkova, Miroslav Mrlik, Marketa Ilcikova, Josef Osicka, Michal Sedlacik, Jaroslav Mosnacek

**Affiliations:** 1Centre of Polymer Systems, University Institute, Tomas Bata University in Zlin, Trida T. Bati 5678, 760 01 Zlin, Czech Republic; ekutalkova@utb.cz (E.K.); marketa.ilcikova@savba.sk (M.I.); osicka@utb.cz (J.O.); msedlacik@utb.cz (M.S.); 2Polymer Institute, Slovak Academy of Sciences, Dubravska cesta 9, 845 41 Bratislava 45, Slovakia; jaroslav.mosnacek@savba.sk; 3Department of Chemistry, Lodz University of Technology, Institute of Polymer and Dye Technology, 90 924 Lodz, Poland; 4Department of Polymer Engneering, Faculty of Technology, Tomas Bata University in Zlin, Vavreckova 275, 762 72 Zlin, Czech Republic

**Keywords:** graphene oxide, (2-(trimethylsilyloxy)ethyl methacrylate), electrorheology, reduction, SI-ATRP

## Abstract

In this study, a verified process of the “grafting from” approach using surface initiated atom transfer radical polymerization was applied for the modification of a graphene oxide (GO) surface. This approach provides simultaneous grafting of poly(2-(trimethylsilyloxy)ethyl methacrylate) (PHEMATMS) chains and a controllable reduction of the GO surface. This allows the fine tuning of its electrical conductivity, which is a crucial parameter for applications of such hybrid composite particles in electrorheological (ER) suspensions. The successful coating was confirmed by transmission electron microscopy and Fourier-transform infrared spectroscopy. The molecular characteristics of PHEMATMS were characterized by gel permeation chromatography. ER performance was elucidated using a rotational rheometer under various electric field strengths and a dielectric spectroscopy to demonstrate the direct impact of both the relaxation time and dielectric relaxation strength on the ER effectivity. Enhanced compatibility between the silicone oil and polymer-modified GO particles was investigated using contact angle measurements and visual sedimentation stability determination. It was clearly proven that the modification of the GO surface improved the ER capability of the system due to the tunable conductivity during the surface-initiated atom transfer radical polymerization (SI-ATRP) process and the enhanced compatibility of the GO particles, modified by polymer containing silyl structures, with silicone oil. These unique ER properties of this system appear very promising for future applications in the design of ER suspensions.

## 1. Introduction

Electrorheological (ER) fluids are smart materials that can change their mechanical properties upon the application of an external electric field. The ER fluids are generally heterogeneous systems mostly composed of solid particles and an insulating liquid medium [1,2]. After the application of an external electric field, the particles are polarized and start to create internal chain-like structures in the direction of the electric field (Scheme 1). This internal structure development leads to the transition from a liquid to a solid-like state accompanied by a viscosity increase in several orders of magnitude [3,4,5]. 

Graphene, one of the carbon allotropes, is a one atom thick carbon sheet consisting of a two-dimensional honeycomb lattice [6,7]. Due to its unique electrical and chemical properties, this material can be successfully utilized in many industrial applications, e.g., in the fabrication of electronic components, energy storage devices (super capacitors and lithium ion batteries), sensors, or drug delivery devices [8,9,10]. Chemical and physical functionalization of the graphene surface is necessary to achieve compatibility with its surroundings and thus several methods have been developed to improve the process of stabilization and the modification of the graphene [9,11,12]. One of them is the oxidation of the graphene by chemical methods, thus obtaining a graphene oxide (GO) with functional hydroxyl, carbonyl, carboxyl, or epoxy groups. These oxygen-containing groups are very important for the covalent bonding of other molecules [12,13]. The oxidation of graphene, however, leads to a very low electrical conductivity and poor ER performance due to the disruption of the conjugation structure of the graphene. However, for the ER application, the graphene needs to be oxidized to decrease its high conductivity, which, from the application point of view, is not desirable, as there is a risk of creating a short circuit [14,15,16]. Nevertheless, these drawbacks of GO can be overcome by an additional reduction of GO, which is usually performed before or after a suitable surface modification. The modification, consisting of grafting polymer chains onto the surface of GO particles, enables the control of its physicochemical properties [17]. The development of hybrid particles based on the GO and polymer coating can find utilization in various areas of applications, such as memory devices [18], flexible supercapacitors [19], or hydrogels for medical applications [20]. The most important benefit for such hybrid particles is the 2D shape of the GO and polymer coating, which provides excellent compatibility with its surroundings. As we recently demonstrated, a simultaneous reduction and coating process of GO led to both an increase of its conductivity, and a significant improvement of the sedimentation stability and ER performance thanks to its substantial shell, thus enhancing compatibility [3,9,15,21]. Similar results were published by other authors, however, in their study, the GO particles were non-covalently modified by various polymers, such as polyaniline [22] and polypyrrole [23], or covalently by silsesquioxane oligomers [24], and generally two steps were needed to receive the polymer coating and the appropriate conductivity for the ER investigation.

This study deals with the modification of the GO particles achieved via surface-initiated atom transfer radical polymerization (SI-ATRP) of 2-(trimethylsilyloxy)ethyl methacrylate (HEMATMS). GO/poly(2-(trimethylsilyloxy)ethyl methacrylate) hybrids (GO-PHEMATMS) with two different molar masses of polymer were prepared with good control over the polymerization. Improved compatibility of the GO-PHEMATMS with silicone oil as a liquid medium as well as improved sedimentation stability were proven. The influence of conductivity on the ER performance was also demonstrated and a system with a uniquely high yield stress was obtained. Moreover, according to our knowledge, such a modification is a pioneer study, with regards to both the GO modification with silyl-based polymer grafting using the SI-ATPR approach, and its impact on the ER performance. 

## 2. Experimental Part

### 2.1. Materials

Graphite (powder, <20 μm, synthetic) as a precursor of GO; sulfuric acid (H_2_SO_4_, reagent grade, 95–98%), sodium nitrate (NaNO_3_, ACS reagent, ≥99%), potassium permanganate (KMnO_4_, 97%), and hydrogen peroxide (H_2_O_2_, ACS reagent, 29–32 wt.% H_2_O_2_ basis) were used as oxidation agents for the preparation of GO sheets by a modified Hummers method as was described previously [25]. α-Bromoisobutyryl bromide (BiBB, 98%) was used as an initiator for the attachment onto the GO surface in the presence of triethyleneamine (TEA, ≥99%) as a proton trap as described elsewhere [25]. Ethyl α-bromoisobutyrate (EBiB, 98%), *N*,*N*,*N*′,*N*″,*N*″-pentamethyldiethylenetriamine (PMDETA, ≥99%), copper bromide (CuBr, ≥99%), and anisole (99%) were used as a sacrificial initiator, ligand, catalyst, and solvent, respectively, for ATRP of 2-(trimethylsilyloxy)ethyl methacrylate (HEMATMS, 99%). All mentioned chemicals as well as diethyl ether (ACS reagent, anhydrous, ≥99%) were purchased from Sigma Aldrich (St. Louis, MO, USA) and used as received except HEMATMS, which was passed through a short basic alumina column to remove a stabilizer. A mixture of polydimethyl siloxane (PDMS), Silgard 184, from Dow Corning (Midland, MI, USA) and dried silicone oil M200 from Lukosoil (Kolín, Czech Republic) was used as a polymer matrix. Tetrahydrofuran (THF, p.a.), dimethyl formamide (DMF, p.a.), acetone (p.a.), ethanol (absolute anhydrous, p.a.), toluene (p.a.), and hydrochloric acid (HCl, 35%, p.a.) were obtained from Penta Labs (Prague, Czech Republic) and used as received except THF, which was dried using flakes of sodium (99.9%). Deionized water (DW) was used during all experiments.

### 2.2. Surface Initiated ATRP

The GO particles were prepared according to previously published papers [26]. Further, initiator immobilization was also performed according to our preceding study [25]. The immobilization onto the surface of GO was performed according to Scheme 2. The esterification reaction between the OH groups of GO and double-functional BiBB in the presence of triethylamine was carried out to obtain the initiator moiety covalently bonded on the surface of GO. The final SI-ATRP grafting procedure is also schematically shown in Scheme 2 and was performed as follows:

GO sheets with an attached ATRP initiator (1 g) were put into a Schlenk flask and the flask was evacuated and backfilled with argon three times. HEMATMS (146.6 mmol, 32 mL), EBiB (1.466 mmol, 0.215 mL), PMDETA (5.864 mmol, 1.22 mL), and anisole (32 mL) were pre-purged with argon, at least 10 minutes each, and added into the flask under argon flow. The system was degassed by three freeze-pump-thaw cycles, filled with argon, and CuBr (1.466 mmol, 0.2103 g) was added to the frozen system under gentle argon flow. The molar ratio of the reactants [HEMATMS]:[EBiB]:[CuBr]:[PMDETA] was either [100]:[1]:[1]:[2] or [200]:[1]:[1]:[4] in order to receive the particles modified with PHEMATMS of the two different chain-lengths (marked later as GO-PHEMATMS-1 and GO-PHEMATMS-2, respectively) and differing also in electrical conductivity. Anisole was used as a solvent in a content of 50 vol.%. The flask with the polymerization mixture was immersed into a 70 °C preheated silicone oil bath to initiate the polymerization process and the polymerization was performed under stirring at this temperature for 2.5 and 5 hours, respectively. Finally, the polymerization mixture was exposed to air to stop the polymerization. The conversions determined by proton spectra of nuclear magnetic resonance (^1^H NMR) spectroscopy (Bruker, Japan, 400 MHz) in CDCl_3_ were 67% and 60% for GO-PHEMATMS-1 and GO-PHEMATMS-2, respectively. The products, GO-PHEMATMS-1 and GO-PHEMATMS-2, were filtered, the particles in the form of filtration cake were redispersed in DMF (200 mL), and washed by acetone (200 mL) through the poly(tetrafluoro ethylene) PTFE filter paper with 0.44 μm. The redispersion and washing was performed twice and final cleaning and drying was done at the same time using diethyl ether (when the filtration cake was washed twice by 100 mL). The molar mass and dispersity of the PHEMATMS chains were determined from the PHEMATMS initiated from the sacrificial initiator and present in the filtrate using GPC (Agilent, Tokyo, Japan) in THF at a flow rate of 1.0 mL min^−1^ using polystyrene standards. 

### 2.3. Particle Characterization

Transmission electron microscopy (TEM) was performed using a JEM-2100Plus microscope (Jeol, Peabody, MA, USA) to evaluate the thickness and uniformity of the polymeric shell. Fourier transform infrared (FTIR) spectra were recorded on a Nicolet 6700 (Nicolet, USA) using an attenuated total reflectance (ATR) technique with Ge crystal and 64 scans with a resolution of 4 cm^−1^ within a wavenumber range of 4000–600 cm^−1^. The Raman spectra (3 scans, resolution of 2 cm^−1^) were collected on a Nicolet DXR (Nicolet, Midland, ON, Canada) using an excitation wavelength of 532 nm. The integration time was 30 s, while the laser power on the surface was set to 1 mW. The powders were compressed to the form of pellets (diameter of 13 mm, thickness of 1 mm) by using a laboratory hydraulic press (Trystom Olomouc, H-62, Olomouc, Czech Republic). The electrical conductivity, *σ*_p_, measurements were performed on the same pellets as were used for Raman investigation, and measured using a Keithley 6517B (Keithley, Austin, TX, USA) multimeter and the four-point method. The contact angle measurement (CA) was evaluated from the static sessile drop method carried out on a Surface Energy Evaluation system equipped with a CCD camera (Advex Instruments, Brno, Czech Republic).

X-ray photo-electron spectroscopy (XPS) measurements were performed using a target factor analysis (TFA) XPS device from Physical Electronics. The base pressure in the XPS analysis chamber was approximately 6 × 10^−8^ Pa. The samples were excited by X-rays over a 400 µm^2^ spot area with monochromatic Al Kα_1,2_ radiation at 1486.6 eV. Photoelectrons were detected with a hemispherical analyser positioned at an angle of 45° with respect to the normal to the sample surface. The energy resolution was approximately 0.5 eV. Survey-scan spectra were acquired at a pass energy of 187.85 eV, whereas for C 1s, individual high-resolution spectra were taken at a pass energy of 29.35 eV and with a 0.125 eV energy step. All the spectra were referenced to the main C 1s peak of the carbon atoms, which was assigned a value of 284.8 eV. The spectra were analysed using MultiPak v8.1c software (Ulvac-Phi Inc., Kanagawa, Japan, 2006) from Physical Electronics, which was supplied with the spectrometer. C1s spectra were fitted with a symmetrical Gauss-Lorentz function. A Shirley-type background subtraction was used.

### 2.4. Preparation of ER Fluids

The corresponding powders (neat GO, GO-PHEMATMS-1, and GO-PHEMATMS-2) were sieved on sieves with a pore diameter of 45 μm to obtain some representative portion for rheological investigation followed by drying in a vacuum oven at 60 °C for 24 h to eliminate potential water residues. The ER fluids were prepared by dispersing the corresponding particles in silicone oil (Lukosiol M200, Chemical Works Kolín, Kolín, Czech Republic; viscosity of *η*_c_ = 194 mPa·s, conductivity of *σ*_c_ ≈ 10^−11^ S·cm^−1^) in a 5 wt.% concentration. Before each experiment, the as-prepared system was first thoroughly stirred with a glass stick for approximately 5 min and then sonicated for 1 min to ensure the homogeneous distribution of the particles within the system.

### 2.5. Electrorheological Measurements

To prove the ER activity of particles under investigation, the steady shear measurements in a controlled shear rate mode were performed using a Bohlin CVOR 150 rotational rheometer (Malvern Instruments, Worchestershire, UK) with a parallel-plate geometry (diameter of 40 mm, gap of 0.5 mm). The external electric fields of a 0.5–2.5 kV·mm^−1^ strength were generated using a DC TREEK 668B high-voltage source (TREK, USA). The experiment protocol was as follows: (i) Shearing the systems for 60 s at a shear rate of 50 s^−1^ either to homogeneously distribute the particles after the gap filling or to destroy residual chain-like structures after the previous measurement at a non-zero electric field strength, *E*; (ii) application of the external electric field of a given strength for 60 s before shearing to provide sufficient time for particles to organize themselves into stable, internal structures within the system; (iii) the shearing stage in the shear rate range of 0.1–300 s^−1^ with 5 points/decade in a logarithmic scaling. On/off cycle measurements were performed at a shear rate of 0.1 s^−1^ and 1.5 kV mm^−1^. One on/off cycle lasts 600 s.

### 2.6. Dielectric Measurements

Dielectric relaxation spectroscopy of the corresponding ER fluids using a high-precision impedance analyser Novocontrol Concept 51 (Novocontrol, Montabaur, Germany) in a broad frequency range of 5–10^5^ Hz was applied as another method for the evaluation of ER activity. As the relaxation mechanism of ER fluids can be affected by temperature [27,28], it should be noted that the characterization was performed at 25 °C.

### 2.7. Sedimentation STABILITY

The sedimentation stability measurement reflecting particles’ compatibility with the carrier liquid was investigated for 5 wt.% ER fluids via visual observation and evaluated by applying the sedimentation ratio concept (defined as the height of the particle-rich phase relative to the total system height), and its development in time.

## 3. Results and Discussion

The presence of PHEMATMS chains onto GO particles was confirmed via several independent techniques. First, the morphological aspect of the polymer grafted onto the GO particles was investigated using TEM images (Figure 1). This analysis evidently confirmed the 2D shape and well-exfoliated structure of individual GO sheets (Figure 1a). After the grafting of PHEMATMS with molar masses of 12 600 g/mol and 20 400 g/mol for GO-PHEMATMS-1 and GO-PHEMATMS-2 (Figure 2), with a dispersity of 1.19 and 1.13, respectively, from the GO surface, the sheet-like structure of GO was retained and the polymer formed a compact layer onto the GO surface (Figure 1b). A darker shade was observed for GO-PHEMATMS-2, i.e., for GO modified with higher molar mass PHEMATMS chains (Figure 1c).

The chemical aspect of GO particles grafted with PHEMATMS chains was monitored using FTIR analysis (Figure 3). As expected, the FTIR spectrum of the original GO is in good agreement with those from previous works: A peak from the epoxy groups at around 823 cm^−1^, a broad peak from the OH groups at around 3500 cm^−1^, and a peak from the carboxyl and carbonyl groups at around 1700 cm^−1^ are all clearly visible, confirming the successful oxidation of graphite powder [25]. The successful grafting by PHEMATMS was confirmed by observation of new absorption bands corresponding to the PHEMATMS. A characteristic sharp peak at around 1726 cm^−1^ reflects the stretching vibration of –C=O from the ester groups, and the broad absorption peak composed of individual sub-peaks at wavenumbers of around 1253 cm^−1^, 1153 cm^−1^, and 1069 cm^−1^ indicates the presence of –Si–CH_3_, –C–O–, and –Si–O– bonds, respectively.

The surface initiated ATRP did not only provide grafting of GO particles with PHEMATMS chains, but also resulted in an increasing electric conductivity, *σ*_p_. While the *σ*_p_ for the original unmodified GO particles was only 1 × 10^−8^ S·cm^−1^, i.e., close to the generally accepted limit of *σ*_p_ for sufficient ER activity [29], the *σ*_p_ of 4 × 10^−7^ and 6 × 10^−6^ S·cm^−1^ were determined for GO-PHEMATMS-1 and GO-PHEMATMS-2, respectively. This is in good agreement with our recent works, where we showed that controllably performed ATRP can provide systems with grafted polymer layers [26] with controlled thicknesses and precisely tuned *σ*_p_ [9].

To further prove that the partial and controllable reduction was performed, the Raman spectra of neat GO and GO-PHEMATMS-1 and GO-PHEMATMS-2 are shown in Figure 4. It can be seen that the ratio between the sp^3^ hybridization (D)/ sp^2^ hybridization (G) peaks’ intensities (I_D_/I_G_) obtained after the SI-ATRP process increases as a confirmation of the partial reduction. Thus, a change from 0.90 to 1.08 and 1.11 was observed after grafting of PHEMATMS-1 and PHEMATMS-2. It is also visible that the 2D sheet-like morphology of the GO created during oxidation remained nearly the same even after modification with various polymer chain lengths.

The presence of the successful modification of the GO surface with PHEMATMS chains is confirmed by the appearance of new binding energies, the Si 2p and Si 2s peaks, from silyl-based monomer units. Moreover, the partial reduction of GO-polymer hybrids was confirmed by XPS spectra (Figure 5), where the carbon (C)/ oxygen (O) ratio increased, and reached values of 2.00, 2.26, and 2.65 for neat GO, GO-PHEMATMS-1, and GO-PHEMATMS-2, respectively, indicating a lower amount of oxygen containing groups on the GO surface. Furthermore, the partial changes in the hybridization of C1s from sp3 to sp2 proved that the delocalized structure was restored and that the reduction took place. This evaluation is also summarized in Table 1.

The ER performance of 5 wt.% systems based on unmodified GO particles and their modified analogues, GO-PHEMATMS-1 and GO-PHEMATMS-2, differing in both the thickness of grafted PHEMATMS and *σ*_p_, was compared both in the absence and in the presence of an external electric field. As can be seen in Figure 6, the ER fluid based on the unmodified GO particles exhibited a linear log–log plot of the shear stress on the shear rate applied in the absence of the electric field, suggesting Newtonian flow behaviour characteristics. The ER fluids based on the PHEMATMS-grafted GO particles possessed a small divergence from the ideal flow behaviour. This deviation results from the improved interactions between the dispersed particles and carrier liquid, reflected in the improved sedimentation stability as will be discussed later. Briefly, the improvement in the mentioned interactions is caused by the decreased surface free energy of GO particles after their grafting with PHEMATMS chains. The particles with high surface free energy generally exhibit a lyophobic character, resulting in their aggregation into clusters in a time longer than that used in the performed rheological experiments. On the contrary, the particles with low surface free energy exhibit a lyophilic character, resulting in better wetting with the carrier liquid, similar to those observed in carbonyl iron-silicone oil systems [30]. Nonetheless, the deviation from the linearity in the case of PHEMATMS-grafted GO particles is negligible in the presence of an external electric field.

After the application of an external electric field, the ER phenomenon occurred for all three investigated systems, due to the formation of internal chain-like structures of polarized dispersed particles, reflecting the Bingham-like flow behaviour characteristics represented by the generation of a yield stress, *τ*_y_, defined as the minimal stress required to start the system flow again. Moreover, the value of *τ*_y_ increased with the increase of *E*. Evidently from Figure 6, the ER effect (defined here as an increase in the Bingham-like flow behaviour at an applied *E*) increased with the grafting of GO particles with PHEMATMS chains and further with the thickness of the polymer layer. This can be a consequence of the enhancement of the *σ*_p_ and/or of the particles’ polarizability as both contribute to the enhanced electro-responsive capabilities of the investigated systems. Moreover, the GO-PHEMATMS-2 based system (Figure 6c) exhibited values of shear stress of nearly 200 Pa at 3 kV/mm, which is the highest value received for GO-based systems up to this date at such low particle concentrations (the following references summarize the GO-based systems and their ER performance [22,31,32,33,34]).

As mentioned above, the stiffness of the internal chain-like structures formed under the external electric field can be proportionally expressed in the *τ*_y_ value. Hence, Figure 7 depicts the dependence of *τ*_y_ on the applied electric field to better illustrate the ER performance of the ER fluids under investigation. The *τ*_y_ values evaluated here were taken as shear stress values at very low shear rate (0.1 s^−1^) in Figure 6. It was proposed in electrorheology that this log–log dependence obeys the power law:
*τ*_y_ = *q* · *E*^α^(1)
where parameter *q* represents the rigidity of the internal chain-like structures formed upon the applied electric field, and the value of parameter *α* should be around 1.5 or 2 for well-developed structures [35]. Evidently, from Table 2, all the investigated ER fluids had an *α* parameter of around 1.5. This, according to the theory, indicates that the conduction model is dominant factor for the ER performance of all three types of dispersed particles used within ER fluids [36]. Furthermore, the parameter, *q*, representing the rigidity of the internal chain-like structures, follows the dependence discussed in the ER effect evaluation part, with a significant increase of the rigidity when the polymer shell thickness is increased.

To prove the reproducibility of the phenomenon, five on/off cycles in which an electric field of 1 kV mm^−1^ was applied were performed (Figure 8). It can be clearly seen that the neat GO based suspensions show slight fluctuations during the on-state cycle and reach values of shear stress around 30 Pa, indicating that the internal structure is not fully developed. However, both the suspensions based on GO-PHEMATMS show a stable behaviour during the on-state regime as a consequence of the well-developed internal structures. This behaviour is also supported by the fact that the shear stresses are 60 Pa and nearly 100 Pa, for GO-PHEMATMS-1 and GO-PHEMATMS-2, respectively, when an electric field of 1.5 kV mm^−1^ was applied.

The suitability of the particles involved in this study for ER application was further confirmed via dielectric spectroscopy measurement. The ER effect generally stems from the particles’ interfacial polarization in the presence of the external electric field. From an investigation of the registered relaxations in the dielectric spectra (Figure 9), the relaxation time, *t*_rel_, and dielectric relaxation strength, Δ*M’*, can be obtained using the well-known Havriliak–Negami model [33] in its modified form [37]:(2)$$M_{HN}^{*}(\omega )={M^{\prime }}_{\infty }+\frac{\Delta M^{\prime }}{{\left(1+{\left(i\omega \cdot t_{rel}\right)}^{\alpha }\right)}^{\beta }}$$
(3)$$M^{*}=M^{\prime }-iM^{\prime \prime }$$
where *M*^*^ is the complex dielectric modulus, *M’*_∞_ is the unrelaxed (high frequency) dielectric modulus value, *i* is the complex unit, and *M*’ and *M*’’ are the elastic and loss dielectric modulus, respectively. The *ω* is the angular frequency (=2*π**f*).

As can be seen in Table 3, both parameters relevant for the efficient ER effect have the expected values. Indeed, *t*_rel_ (located for all the investigated ER fluids within the frequency range of interfacial polarization) decreased, causing a faster response by the application of the external electric field, and the dielectric relaxation strength increased with the grafting of GO particles and with the higher molar mass of the polymer, due to both the better compatibility and the enhanced electric conductivity, and finally with the higher molar mass of the polymer. Therefore, the dielectric relaxation spectroscopy absolutely confirmed the results obtained in the ER experiments.

The grafting of GO particles with PHEMATMS also positively influenced the stability of the investigated ER fluids against sedimentation. Both PHEMATMS-grafted GO systems exhibited a significantly higher sedimentation stability, expressed as the sedimentation ratio vs time dependence, than that based on the unmodified GO particles (Figure 10). The stability was more pronounced for GO-PHEMATMS-2, i.e., GO grafted with longer polymer chains (higher thickness of the polymer shell) probably due to the more lyophilic character of these particles causing their repulsion, which, as a result, eliminates particles aggregation. Moreover, as can be seen from the Figure 10 inset, the contact angle measurements showed that the mentioned grafting also influenced the wettability with contact angle values of 49.9° ± 3.2°, 26.3° ± 2.7°, and 24.9° ± 1.4°, for neat GO, GO-PHEMATMS-1, and GO-PHEMATMS-2, respectively. All these findings significantly confirm the enhanced compatibility between the GO particles and the carrier liquid.

## 4. Conclusions

A simple, single-step method of simultaneous grafting of the polymer layer and increasing the electric conductivity of the low conducting GO particles was used to prepare PHEMATMS-modified GO. The successful grafting with the silyl-based polymer was confirmed by TEM and FTIR spectroscopy. The parameters of the PHEMATMS chains were elucidated using the GPC and NMR technique, showing molar masses 12 600 g/mol and 20 400 g/mol for GO-PHEMATMS-1 and GO-PHEMATMS-2, respectively, while narrow polydispersity indexes of 1.19 and 1.13, respectively, were obtained. Electrorheological performance was significantly influenced by the mentioned grafting. The yield stress of 200 Pa, determined for the system containing GO-PHEMATMS-2 at 2.5 kV mm^−1^, is uniquely high and has not been seen for GO-based systems at such low particle concentrations so far. The conductivity mechanism of the chain-like structure formation was consistent with previously published ones and the dielectric properties, together with the enhanced particle conductivity, significantly contributed to the improved ER capability. The reproducibility of the phenomenon was investigated by fiveon/off cycles and it was shown that GO-PHEMATMS based suspensions do not exhibit any fluctuation during the on-state and show shear stresses of 60 Pa and 100 Pa at 1.5 kV mm^−1^ for GO-PHEMATMS-1 and GO-PHEMATMS-2, respectively. Finally, based on the contact angle results between the particles and the carrier liquid varying from 49.9° for unmodified GO down to 26.3°and 24.9° for PHEMATMS-grafted GOs, it was demonstrated that a more hydrophobic character was obtained and thus the later systems exhibited an enhanced sedimentation stability, and provided a very promising system for various applications in electrorheology.

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
