# Peer review of "Enhanced and Tunable Electrorheological Capability using Surface Initiated Atom Transfer Radical Polymerization Modification with Simultaneous Reduction of the Graphene Oxide by Silyl-Based Polymer Grafting"

_nanomaterials, 2019, doi:10.3390/nano9020308_

Reviewer 1 Report

With proper revision efforts, it can be accepted as is.

Author Response

Please see the attachment. "Response to reviewers"

Reviewer 2 Report

The new version of the manuscript has been improved and thus in my opinion it could be published in "Nanomaterials".
However, in the following I suggest some corrections/improvements.
1) page 2, line 1: "which is crucial" -> "which is a crucial"
2) page 3, line 14: "leads to suppressed" -> "leads to very low" (not to fall into contraddiction, since in the following line you are saying that a reduction of the conductivity is positive in ER applications)
3) page 3, line 14: the value of 10**{-5} S cm**{-1} seems too low for graphene: is it necessary to indicate this value?
4) page 3, line 14: "which s" -> "which is"
5) page 6, line 14: "out to obtained" -> "out to obtain"
6) page 13, line 11: "sustained" -> "remained"
7) page 13, line 12: "variou" -> "various"
8) page 13, last 4 lines: the sentence beginning with "The presence of the" should be rewritten because at the moment its form is not correct and I can not understand the correct meaning.
9) page 14, line 1: "spectra Fig. 5" -> "spectra (Fig. 5)"
10) page 14, line 1: "reaches" -> "reached"
11) page 14, line 5: "renewed" -> "restored"
12) page 14, capture of Fig 5: "articles" -> "particles"
13) page 15, line 8: "in mentioned" -> "in the mentioned"
14) page 15, line 11: "in time longer" -> "in a time longer"
15) page 15, line 11: "in performed" -> "in the performed"
16) page 15, line 12: "he particles" -> "the particles"
17) page 15, line 17: "he ER phenomenon" -> "the ER phenomenon"
18) page 16, line 3: "stress nearly" -> "stress of nearly"
19) page 17, line 3: "field applied to better" -> "field to better"
20) page 17, line 5: "Fig. 7" -> "Fig. 6"
21) page 17, line 9: "around the range of" -> "around"
22) page 17, line 15: "rigidity with" -> "rigidity when"
23) page 20, line 15: the symbol before "is the unrelaxed (high frequency) electric" is wrong
24) page 23, line 2: "has not be seen" -> "has not been seen".

Author Response

Please find the response to reviewers in the attachment.

Reviewer 3 Report

The authors have been addressed all the reviewers' concerns for improving their work, thus would suggest for publication of the work.

Author Response

(The authors gave the same response as above.)

Reviewer 4 Report

An improvement in the manuscript can be observed. Many of the directions were attended. However, some English language and style need to be checked. 

Author Response

Please find the response to reviewers comments in the attachment.

Round  2

Reviewer 2 Report

In my opinion, now the paper can be accepted for publication in "Nanomaterials".
I only suggest 4 minor changes:
1) page 4, line 3: "significantly improvement" -> "significant improvement"
2) page 8, line 11: "by using of a laboratory" -> "by using a laboratory"
3) page 11, caption of Figure 2: "over the time" -> "versus the time"
4) page 23, line 8 of the aknowledgements: "thanks to National" -> "thanks National"

Author Response

1) page 4, line 3: "significantly improvement" -> "significant improvement"

Based on the corrections by English native speaker the changes were performed accordingly.

2) page 8, line 11: "by using of a laboratory" -> "by using a laboratory"

Please see the revised manuscript
3) page 11, caption of Figure 2: "over the time" -> "versus the time"

Please see the revised manuscript
4) page 23, line 8 of the aknowledgements: "thanks to National" -> "thanks National"

Please see the revised manuscript

Reviewer 4 Report

In general, the quality of the paper was significantly improved and all the comments were addressed, but the English edition is still needed. 

Author Response

The English language has been carefully checked by English native speaker to improve the English quality of the article (Please see the revised manuscript).

This manuscript is a resubmission of an earlier submission. The following is a list of the peer review reports and author responses from that submission.

Round  1

Reviewer 1 Report

Using surface initiated atom transfer radical polymerization (SI-ATRP), authors modified graphene oxide (GO) surface with poly(2-(trimethylsilyloxy)ethyl methacrylate) (PHEMATMS) chains, resulting in controllable reduction of GO surface with fine tuning of its electrical conductivity. It was clearly observed that presented modification of GO surface improved the ER capability of the system due to the tunable conductivity during SI-ATRP process and enhanced compatibility of GO particles, modified by polymer containing silyl structures, with silicone oil. The modification of GO is very novel and interesting, insuring its publication with a following minor comment.

Please note that the electrical conductivity of GO prepared from the Hummers method is often to be considered to be high for the rheological measurement depending on the choice of rheometer. Therefore instead of increasing the electrical conductivity of the GO, various reports on how to decrease the conductivity of GO have been attempted. This issue needs to be added while the purpose of this work is to increase the conductivity of the GO for the ER. Please see “ACS Appl. Mater. Interfaces 4, 2267 (2012)”, “Soft Matter 10, 6601 (2014)”, “Chem. Rev. 39, 228 (2010)”.

Reviewer 2 Report

Article: “Enhanced and tunable electrorheological capability using SI-ATRP modification with simultaneous reduction of the graphene oxide by silyl-based polymer grafting” By Kutalkova et al.

General comments

The paper is very interesting and presents novelty in the surface initiated ATRP fied to produce after ATRP modification, ER fluids more stable to sedimentation. However, significant improving has to be performed, for example in the abstract.

The abstract does not present introduction and the impact in terms of application and future trends. It is also very important to include a very careful edition of the English used in the paper because there are several grammar mistakes.

In the abstract, (see line 15) “This approach provides simultaneous grafting of the poly(2-(trimethylsilyloxy)ethyl methacrylate) (PHEMATMS) chains and controllable reduction of GO surface with fine tuning of its electrical conductivity, which is crucial parameter for application of the such hybrid composite particles in electrorheological (ER) suspensions…”, in the manuscript I could not evidence the partial reduction of GO in any of the techniques used for characterization. Please clarify.

Normally for the keywords, four to five words (maximum) are used.

In the introduction, it is important to highlight the advantages of SI-ATRP to produce hybrid materials. Also, the authors should include the application of the modified materials with tunable properties in different areas of life and in the industry. The impact should be clarifying to understand the real impact of producing this kind of materials, not only by the synthetic approach but in real life.

Line 38 – “After the application of external electric field, the particles are polarized and start to create the internal chain-like structures in the direction of the electric field” – It will be very helpful to include a scheme for this behavior to clarify the idea for the readers.

Line 64 – “This study deals with the modification of the GO particles achieved via the surface-initiated atom transfer radical polymerization (SI-ATRP) technique” It is very important to define the novelty of this study and highlight it at the end of this section.

Line 67 – The authors claim “thanks to good control over the polymerization”. There are no studies about this in the manuscript, there are no PDI or molecular weight analysis over time to determine if is a really controlled process. The authors should clarify.

Experimental part

It is very important to determine the manufacturer, city, state and country of the reagents and equipment. Please review all the reagents and equipment and complete this information. Please specify the characteristics of the alumina (degree, particle size, etc.). Also specify the characteristics of the deionized water and the equipment to prepare it.

In the surface initiated ATRP conditions, please summarize the way how the initiator was obtained, even the original procedure is cited, because some clarification of the procedure is really needed here.

In line 113, please specify what kind of paper filter was used for the filtration of the final products. Line 113 please correct “disersed” by “dispersed”. Line 116 please add a point after “100 mL”.

I never verified the GPC analysis during the manuscript, I think this should be included for both GO

PHEMATMS samples.

Results and discussion

Line 170 – The expression “This analysis evidently confirmed the 2D shape and well-exfoliated structure of individual GO sheets (Fig. 1a)”, it needs more explanation, because this is the reason why graphene oxide formation is confirmed.

Line 173 – The expression “from the GO surface the sheet-like structure of GO retained” should be “from the GO surface the sheet-like structure of GO was retained”

Line 185 – “The successful grafting by PHEMATMS was confirmed by observation of new absorption bands corresponding to the PHEMATMS polymer…” I think this is a good evidence, but the grafting modification needs a stronger evidence, which could be using XPS technique, for example, demonstrating the presence of the new binding energies. The FTIR by itself does not probe that because it could be a result of a surface adsorption only, not a covalent bond as authors claim.

Line 193 – I strongly recommend to include a deeper explanation for the increasing in conductivity in terms of the polymer structure.

Line 278 – increase the quality of the equation (B).

Line 298 – “The stability was more pronounced for GO-PHEMATMS-2, i.e. GO grafted with longer polymer chains (higher 299 thickness of the polymer shell)…” what about the effect of polymer architectures in the stability, will also have an effect?

Line 303 – “contact angle values of 49.9°, 26.3°and 24.9°…” what were the SD of the measurements?

General comments:

It will be very interesting to include a grafting ratio effect onto the ER fluid properties studied and also solvent effect.

Conclusions

Line 325 – “Finally, due to the significantly decreased contact angle between…” I do not think is because of the contact angle values, I think is more related to hydrophobicity changes instead.

Reviewer 3 Report

E. Kutalkova et al. were reported a practical research work on silyl-based polymer grafting graphene oxide by using ATR polymerization method. The PHEMATMS-modified GO show enhancing rheological properties as well as gentle reduction of GO. However, I would not recommend for publication of the work in Nanomaterials.

The work could be considered again after significant improvement, unless otherwise might be considered by other specific MDPI journal.

Comments:

Characterization to prove the ‘grafting from’ of PHEMATMS on GO is not enough by using only FTIR, and TEM image is not a convincing method to confirm covalent bonding.

The reduction level is very slightly and the author did not explain why?

Similarly the enhancing rheological properties are not explained why and how?

Reviewer 4 Report

The paper reports a study of the electrorheological properties of graphene-oxide particles modified by grafting polymer chains (in particular PHEMATMS) on their surface and simultaneous reduction. The manuscript describes the details of the performed experimental activity and the achieved results. In particular, the resulting material has good electrorheological properties, with improved compatibility with silicone oil and enhanced sedimentation stability.
In my opinion, the manuscript is scientifically interesting and could be useful for a possible electrorheological application of graphene-oxide based particles.
However, some clarifications and corrections should be made to the paper before a possible pubblication (I am also attaching a pdf file listing some modifications I suggest).
In particular:
1) the authors should spend a few words on the possible utility of the use of graphene-oxide based particles in electrorheological applications, compared to other possible materials;
2) when, in the introduction (lines 42-44), the Authors talk about graphene, I suggest to add the following two references:
A. H. Castro Neto, F. Guinea, N. M. R. Peres, K. S. Novoselov, and A. K. Geim, "The electronic properties of graphene", Rev. Mod. Phys. 81, 109 (2009), DOI: 10.1103/RevModPhys.81.109
P. Marconcini and M. Macucci, "The k.p method and its application to graphene, carbon nanotubes and graphene nanoribbons: the Dirac equation", La Rivista del Nuovo Cimento 34, Issue N. 8-9, 489-584 (2011), DOI: 10.1393/ncr/i2011-10068-1
3) I would not use acronyms in the title and in the abstract; moreover, in the text some acronyms have not been explained (I have highlighted these points in green in the attached pdf file);
4) in the attached pdf file, I have also written in red several corrections I suggest;
5) in the same pdf file, I have highlighted in yellow some parts that are not clear and thus should be written better. In particular, in the lines 276-284 the relation between M', M" and M* should be clarified.
6) in Fig. 5 the writing "on state" is not visible;
7) in the conclusions (lines 315-316) the Authors talk about "narrow polydispersity indexes of 1.19 and 1.13"; what part of the text they are referring to?
8) in most references, the DOI is attached to the page number.
